# Altered TDP-43 Structure and Function: Key Insights into Aberrant RNA, Mitochondrial, and Cellular and Systemic Metabolism in Amyotrophic Lateral Sclerosis

**DOI:** 10.3390/metabo12080709

**Published:** 2022-07-29

**Authors:** Leanne Jiang, Shyuan T. Ngo

**Affiliations:** 1Australian Institute for Bioengineering and Nanotechnology, The University of Queensland, Brisbane, QLD 4072, Australia; 2Perron Institute for Neurological and Translational Science, Perth, WA 6009, Australia; 3School of Biological Science, The University of Western Australia, Perth, WA 6009, Australia; 4Centre for Clinical Research, The University of Queensland, Brisbane, QLD 4029, Australia; 5Department of Neurology, Royal Brisbane and Women’s Hospital, Brisbane, QLD 4029, Australia

**Keywords:** amyotrophic lateral sclerosis, ALS, TDP-43, RNA, autoregulation, splicing, non-coding, OXPHOS, mitochondria, mitochondrial dynamics, metabolism, lipid metabolism

## Abstract

Amyotrophic lateral sclerosis (ALS) is a progressive and fatal neuromuscular disorder with no cure available and limited treatment options. ALS is a highly heterogeneous disease, whereby patients present with vastly different phenotypes. Despite this heterogeneity, over 97% of patients will exhibit pathological TAR-DNA binding protein-43 (TDP-43) cytoplasmic inclusions. TDP-43 is a ubiquitously expressed RNA binding protein with the capacity to bind over 6000 RNA and DNA targets—particularly those involved in RNA, mitochondrial, and lipid metabolism. Here, we review the unique structure and function of TDP-43 and its role in affecting the aforementioned metabolic processes in ALS. Considering evidence published specifically in TDP-43-relevant in vitro, in vivo, and ex vivo models we posit that TDP-43 acts in a positive feedback loop with mRNA transcription/translation, stress granules, cytoplasmic aggregates, and mitochondrial proteins causing a relentless cycle of disease-like pathology eventuating in neuronal toxicity. Given its undeniable presence in ALS pathology, TDP-43 presents as a promising target for mechanistic disease modelling and future therapeutic investigations.

## 1. Introduction

Amyotrophic lateral sclerosis (ALS) is a progressive and fatal neuromuscular disorder, causing degeneration of upper and lower motor neurons in the cortex and spinal cord [1]. The majority of patients are classified as having sporadic ALS (sALS) with no known family history of the disease nor an inherited mutation. However, patients with familial ALS (fALS) account for ~10% of cases; fALS patients have a known family history of ALS and/or harbour an inherited mutation in known disease-causing genes such as: *SOD1*, *C9orf72*, *NEK1*, *ATXN2*, and/or *TARDBP* [2]. ALS is regarded as a highly complex and heterogeneous disease; clinical features vary greatly from patient to patient and even those carrying the same rare genetic mutation can present with drastically different symptoms. In 2006, two seminal findings identified hyper-phosphorylated and ubiquitinated cytoplasmic TAR-DNA binding protein-43 (TDP-43) inclusions within the anterior horn motor neurons in ~97% of ALS patients—a stark contrast to the ~1% carrying a mutation in *TARDBP* [3,4]. The presence of these hyper-phosphorylated TDP-43 cytoplasmic inclusions are now considered a key pathological characteristic of ALS [4,5]. Interestingly, these cytoplasmic inclusions are not restricted to the ventral horn of the spinal cord or the motor regions of the cortex; in fact, other neuronal and glia cells also exhibit these inclusions diffused through areas such as the temporal cortices and the hypothalamus [3,4,6]. This suggests a more systemic role of TDP-43 in ALS disease pathology.

TDP-43 is an essential protein involved in RNA processing [7], cellular metabolism [8], and mitochondrial function [9]. It is widely accepted that metabolic processes (both cellular and systemic) are dysregulated in ALS patients and disease relevant models. The majority of *TARDBP* mutations reside in the hyper-mutable glycine-rich C-terminal domain, with other mutations spread across the different domains such as the RNA recognition motifs (P112H, D169G, K181E) and the nuclear localisation signal (A90V)—with potentially several more mutations yet to be identified [10,11,12] (Figure 1). Several groups have generated models with rare *TARDBP* mutations to understand their contribution towards protein misfolding and neuronal toxicity in ALS. In vitro studies of the C-terminal Q331K and G294A mutations identified a higher propensity to form C-terminal fragments, leading to cell death; most often these mutations produced new phosphorylation sites or disrupted RNA binding sites [13]. By contrast, the NLS domain A90V mutation results in cytoplasmic inclusions, however does not cause cell death in vitro [14]. These studies suggest that although TDP-43 mutations can cause cytoplasmic aggregates, additional factors may be required for neuronal toxicity and subsequent motor neuron death; in line with the multi-step hypothesis of ALS [15]. Given the large number of TDP-43 RNA targets, a disruption of normal TDP-43 could lead to disease-causing mechanisms such as RNA instability, oxidative stress, mitochondrial dysfunction, and even dysregulated lipid metabolism.

## 2. Normal Physiological TDP-43 Structure and Function

The *TAR-DNA binding protein* (*TARDBP*) gene encodes a highly conserved and ubiquitously expressed 43 kDa heterogenous nuclear ribonucleoprotein (hnRNP), TDP-43. *TARDBP* is located on chromosome 1 and comprises of 6 exons forming 414 amino acids (aa). These six exons form domains that are bookended by an N-terminal domain (NTD) (aa1–76) and C-terminal domain (CTD) (aa274–414). Like many other hnRNPs, the domains within TDP-43 include a nuclear localisation signal (NLS aa82–98), two RNA recognition motifs (RRM1 aa106–176; RRM2 aa191–259), and a nuclear export signal (NES aa239–250) residing within RRM2 (Figure 1) [16,17]. Additionally, specific sites are located within these domains for other functions such as caspase-3 cleavage sites, post-translational modification, and mitochondrial localisation (reviewed in Buratti [18]). Ultimately, the unique structure of TDP-43 gives rise to its capacity to bind and regulate over 6000 RNA species, in addition to its own autoregulation abilities. The specific functions of TDP-43 include mRNA stabilisation [19], transcription [20,21], translation [22], splicing [21,22,23], axonal transport [24], apoptosis [23], microRNA processing [25], epigenetic modifications [26] and cryptic exon inclusion/repression [27,28]. It is without doubt that any alteration to these structural domains and/or normal functions of TDP-43 could cause downstream destabilisation effects of the intended target(s).

### 2.1. The N-Terminal Domain

The main functions of the NTD are to self-regulate *TARDBP* mRNA through homodimerisation and aid mRNA splicing [29,30,31,32,33]. TDP-43 can form homodimers, spanning a spectrum of oligomeric species with dynamic folding states [31,34]. NTD generated TDP-43 oligomers do not interact with other RNA like proteins such as hnRNPA1/2 and are distinct from stress or inclusion formed cytoplasmic oligomers [30,31]. It is speculated that NTD generated oligomers adopt structures resistant to cellular stress and act as preventative measures against cytoplasmic inclusion formation under normal conditions [32,35]. However, in a stressed HEK293T TDP-43 overexpression model, these NTD oligomers were aggregated within large cytoplasmic inclusions, apparently yielding to cellular stress [29]. Despite this, under physiological conditions there is debate surrounding the importance and function of these oligomers. Avendaño-Vázquez et al., has suggested that a dimer equilibrium is required for proper TDP-43 function [20]. TDP-43 protein missing the NTD displays significant β-sheet folding abnormalities, consequently leading to a loss of TDP-43 function [30,32]. Interestingly plasmids missing just the first 10 amino acid residues have been shown to exhibit similar significant β-sheet folding abnormalities and structural destabilisation, suggesting the first 10 amino acids are the most crucial of the 76 in the NTD [30,32]. Additionally, the NTD of TDP-43 is known to enhance exon 9 inclusion of the *CFTR* gene; should exon 9 be excluded, patients develop cystic fibrosis [28]. In several in vitro models inducing NTD mutations, TDP-43 was unable to repress aberrant splicing of exon 9 in *CFTR*, indicating the importance of the NTD in regulating splicing [30,32,33]. In another in vitro study, point mutations at these NTD sites: V31R, T32R, and L28A resulted in similar incomplete TDP-43 folding, abolished splicing activity, as well as induced cytoplasmic inclusions [36]. Taken together, these studies demonstrate the importance of NTD oligomers for proper folding, splicing, RNA metabolism, and in preventing the formation of cytoplasmic aggregates [31,34].

### 2.2. RNA Recognition Motifs

RRMs direct RNA and DNA binding. TDP-43 RRMs sit in tandem and can bind to over 6000 targets, with a 20–60-fold higher binding affinity for long UG repeats [37,38,39,40]. These RRMs are separated by a unique 15 amino acid cross-linker spanning four β-strands, allowing for additional interactions between TDP-43 and other RNAs [39]. The RRM domains of TDP-43 enable splicing control [41,42], pre-mRNA regulation [43], physiological oligomer formation [44,45], and possibly aggregate clearing [43,44,46]. Although these RRMs are similar in sequence, they form different tertiary structures, where RRM1 preferentially binds longer (UG_6_) repeats, and RRM2 binds shorter (UG_3_) repeats [39]. Interestingly, these RRMs are prone to oxidative stress due to the occurrence of cysteine residues; these residues are accessible on the β-loop of RRM1, however are buried and inaccessible on RRM2 [46]. Due to these structural differences, the RRM2 domain are able to form highly stable assemblies with DNA strands, although it is unknown whether these assemblies contribute to cytoplasmic aggregations [17]. It has been demonstrated that these RRMs induce oligomer formation as a means to prevent TDP-43 aggregation [44]. However, when cysteine residues on RRM1 are oxidised they undergo oligomer conformational changes resulting in TDP-43 insoluble aggregates [46]. In an NSC-34 stressed model, oligomer formation was disrupted, leading to binding with RNAs destined for stress granule formation [47]. In a HEK293T TDP-43^K181E^ mutation model the structure of TDP-43 and its RRMs were unaffected; however phosphorylated and ubiquitinated TDP-43 increased 4-fold [42]. Upon further examination, these phosphorylated TDP-43 also sequestered healthy TDP-43 into their inclusions suggesting a detrimental effect of mutated TDP-43 RRMs [42]. Taken together, there is a scarcity of studies examining the role of RRMs and their effect on RNA metabolism and TDP-43 aggregation. Thus, further research is required to fully elucidate the role of RRMs in RNA metabolism, cellular toxicity, and its propensity to negatively react to oxidative stress, a key mechanism put forth for ALS.

### 2.3. Nuclear Localisation and Export Signals

Shuttling of TDP-43 to and from the nucleus and the cytoplasm is controlled by the NLS and NES domains [48]. Under physiological conditions TDP-43 is predominantly localised to the nucleus and is exported to the cytoplasm for crucial activities such as mRNA stability and binding [48]. In mouse in vitro (3T3) and in vivo models, overexpression of TDP-43^WT^ does not induce cytoplasmic inclusions, however TDP-43 was observed more frequently in the cytoplasm compared to the nucleus [48,49]. Nuclear localisation targeting is dependent on two clusters of basic residues around aa82–98 [50]. When both clusters are mutated, impaired nuclear localisation occurs and extensive accumulation of TDP-43 cytoplasmic C-terminal fragments are observed [32,48,50]. However, when only the first cluster is altered, TDP-43 is still able to localise to the nucleus [48]. Evidently, redundancies are in place for nuclear localisation should one cluster become impaired.

It could also be possible that defective export may cause TDP-43 to remain in the nucleus, prohibiting normal mRNA capabilities within the cytoplasm. Most hnRNPs are exported by a broad cargo exportin (XPO1) [51]. Previously, TDP-43 nuclear magnetic resonance and predictive modelling has been used to visualise binding domains of NES within the RRM [52]. Interestingly, all potential binding sites of the NES are buried within the core of the RRMs, indicating that the NES is inaccessible for XPO1 binding [52]. In line with this finding, when an NES plasmid was exposed to XPO1, only weak bonds formed; demonstrating that the NES may not be the main domain required for shuttling TDP-43 into the cytoplasm [52]. Upon removal of the NES, similar amounts of TDP-43 were observed in the cytoplasm compared to wild-type, suggesting a ‘passive’ movement of TDP-43 to the cytoplasm in the absence of XPO1 binding [52]. However, when inhibiting the NES, TDP-43 forms insoluble inclusions, suggesting that the NES is still required for RRM functions and proper export [48,52]. Despite limited research into the specific structure of TDP-43 NLS and NES domains, it is plausible that shuttling of TDP-43 is also dictated by the self-regulation or splicing of its RRMs and CTD. Other possibilities include other exportins such as exon junction complexes, nuclear export receptors, and THO/transcription export complexes which have not yet been investigated in TDP-43 models. Further investigations are still warranted to fully characterise the NES and NLS domains to understand how they might contribute to the accumulation of disease relevant TDP-43 pathology.

### 2.4. Glycine-Rich C-Terminal Domain

The main function of the glycine rich CTD is to mediate protein-protein interactions via hnRNP interactions [53]. Other known functions include export and shuttling [48,54], gene transcription [55], and splicing regulation [56]. One way in which the glycine rich CTD modifies gene regulation is through the Q/N (asparagine) rich region, also known as the prion-like region [57]. This prion-like region drives gene regulation through RNP granule assembly [58]. These granules play a vital role in the stress granule response, however, should the clearance and reversibility of these granules fail, a high propensity for aggregation occurs [59]. It has been suggested that disruptions to the nucleic-acid-binding capability and the removal of the C-terminal tail causes TDP-43 mislocalisation due to the improper return to the nucleus [48].

The structure, function, and disruption of the CTD has mostly been studied in the context of ALS, due to its propensity to mutate. Most studies focus either on generating C-terminal fragments or introducing point mutations. Cleavage of TDP-43 via caspase3/7 at Asp89 and Asp174 generates 25 kDa or 35 kDa C-terminal fragment and a quickly degraded N-terminal fragment [60,61]. Due to RRM domain retention, these generated C-terminal fragments can bind to one another forming oligomers [39]. A wide scope of evidence in in vitro and in vivo models indicate that neuronal toxicity is caused by C-terminal cytoplasmic inclusions [62,63], 25 kDa fragment overexpression [64], and/or 35 kDa fragment overexpression [65,66,67]. However, the demonstration of neuronal toxicity by these C-terminal fragments are scarce and remain poorly understood. What we do know is that the 35 kDa fragment alters pre-mRNA splicing, sequesters full length TDP-43 at RRM1, and forms oligomer stress granules via TDP-43 recruitment [65,66,67]. On the other hand, it has been shown that the 25 kDa fragment in a HEK293T model induces cell death, despite being unable to alter or bind to full length TDP-43 [64]. It is clear that neuronal toxicity occurs; however, the mechanism is unknown and more importantly remains elusive for individuals lacking *TARDBP* mutations given that the 25 kDa fragment is most commonly identified in post-mortem studies of ALS patients [67,68]. It is likely that a greater understanding of other post-translational modifications involving phosphorylation, caspases, or even oxidative stress may aide in our knowledge of cytoplasmic TDP-43 induced neuronal toxicity in ALS.

## 3. RNA Instability and Metabolism

RNA stability and degradation are a tightly regulated process to undergo normal protein translation and cellular functions. RNA binding proteins—inclusive of TDP-43—are key players in RNA transcription, splicing, transport, stability, and degradation [69,70]. Mutations, abnormal localisation, and aggregation of RNA binding proteins are clear indicators of RNA dysfunction in ALS [71]. In a transcript study of HEK293T TDP-43^A315T^ and TDP-43^M337V^ mutations, two major clusters identified the interactions between TDP-43 with: splicing and translational defects—clearly demonstrating aberrant RNA metabolism with TDP-43 [22]. Pathologically, TDP-43 localisation within cytoplasmic inclusions, also clearly indicate defective RNA regulation in ALS. Intriguingly many of the RNA targets of TDP-43 are involved in synaptic function and neuronal development [72,73]. Despite clear evidence put forth for RNA instability and dysfunction, the cause and timing of RNA failure remains unclear. The unique properties of TDP-43, such as its own autoregulation, phosphorylation sites, its ability to splice other targets, its close relationship with non-coding RNAs, and possible mechanisms in epigenetics highlight a few potential points of failure, which are discussed below.

### 3.1. Autoregulation of TDP-43

Proper TDP-43 regulation and localisation is critical for cell survival. Most notably, knocking out TDP-43 in murine models results in early embryonic death [74]. On the other hand, overexpression of TDP-43 results in severe neurodegenerative pathological hallmarks in murine and cell models [75,76]. Regulation and localisation of TDP-43 is mostly controlled by its own autoregulation via a negative feedback loop [55]. The 3′UTR contains its own 700 nucleotide TDP-43 binding region (TDPBR) [55]. Alternatively spliced variants of the *TARDBP* mRNA is controlled by four main poly-adenylation sites (pA_1_, pA_2_, pA_3_, pA_4_) [55]. In steady-state RNA splicing pA_1_ is the predominant polyadenylation site for mature *TARDBP* mRNA, whereas a smaller *TARDBP* mRNA isoform utilising the pA_4_ site also exists [55]. The two main proposed mechanisms by which autoregulation occurs to control mRNA transcript and protein levels remains debated.

The first proposed mechanism is via elevated levels of nuclear TDP-43 that readily bind to the TDPBR, promoting transcript instability via exosome degradation within the nucleus [55,77]. The second proposed mechanism suggests that an alternatively spliced version of *TARDBP* (isoform V2) excises pA_1_, subsequently inducing a premature termination codon, thus destined for nonsense mediated decay in the cytoplasm [78,79]. However, two studies argue that *TARDBP* V2 cannot be involved in autoregulation due to a lack of transcript fold-change in a HEK293T TDP-43 overexpression model [20,55]. A second argument is made that despite the excision of the pA_1_ site, pA_2_ is utilised which would then retain TDP-43 in the nucleus for nonsense mediated decay [20]. However, in other studies the inhibition of isoform V2 via antisense oligonucleotides increased canonical *TARDBP* mRNA expression in both murine and human-derived in vitro models, thereby supporting the second proposed mechanism of nonsense mediated decay within the cytoplasm [79,80]. Despite contention, both mechanisms cannot be ruled out as studies to date have been conducted in differing models with differing *TARDBP* transcripts. Murine *TARDBP* can transcribe up to 19 distinct alternatively spliced transcripts, whereas only 10 have been identified in human *TARDBP* studies, proving both hypotheses of TDP-43 autoregulation are feasible, and perhaps even other unidentified mechanisms possible [81].

Regardless of the mechanism in which autoregulation occurs, the activity of TDP-43 is dependent on the integrity of the glycine-rich CTD [82]. TDP-43 mutations residing on the CTD lead to the loss TDPBR binding, and consequently, a loss in the ability to control endogenous transcript levels [55]. Any increase in endogenous *TARDBP* mRNA has been shown to induce a loss of nuclear TDP-43 and an increase in cytoplasmic TDP-43 aggregates [79,83]. Loss of nuclear *TARDBP* mRNA and subsequent TDP-43 protein leads to the loss of neurite outgrowth gene *STMN2* [84], synaptic function gene *UNC13A* [85], and chromatin accessibility of long interspersed nuclear elements retrotransposons [86]. These findings suggest that perturbed autoregulation could be one of the causes of TDP-43 dysfunction in the pathogenesis of ALS. However, it should be noted that autoregulation dysfunction should not be viewed as mutually exclusive to other potential mechanisms as described below.

### 3.2. Phosphorylation and Autophagy of TDP-43

The abnormal accumulation of TDP-43 into cytoplasmic aggregates are the hallmark feature of ALS [4]. Given the large number of serine, threonine, and tyrosine residues, TDP-43 is highly prone to phosphorylation (a form of epigenetic modulation, as discussed in Section 3.5). There are many benefits to phosphorylation [87], however abnormal and/or hyper-phosphorylation can lead to detrimental effects on mRNA translation and alternative splicing regulation [88,89]. To determine whether TDP-43 cytoplasmic aggregates are phosphorylated, two groups collapsed these TDP-43 fragments using dephosphorylation methods—the subsequent separation of these fragments into several distinct bands is indicative of hyperphosphorylation [3,4]. Previous findings indeed demonstrated that the CTD of TDP-43 contains multiple serine sites prone to abnormal phosphorylation at Ser379, Ser403, Ser404, Ser409, and Ser410 [90]. Antibodies raised against these sites did not appear to stain nuclear regions, suggestive of disease specific pathology [90]. Interestingly, the propensity for TDP-43 to aggregate decreases significantly when the Ser409/410 site is mutated to an asparagine, suggesting that phosphorylation of TDP-43 is a driver of cytoplasmic aggregations—but that, however, is not an absolute requirement [91].

Several studies have used multiple models to identify the kinases responsible for TDP-43 phosphorylation. Of those, casein kinase-1 has been shown to phosphorylate many serine residues—in particular the Ser409/410 site—in vitro [90,92,93] and in a drosophila model [94]. Other kinases with proposed phosphorylation capacity include casein kinase-2 and GSK3β; however, little is known about their interactions and impact on TDP-43 phosphorylation [90]. In a drosophila model, casein kinase-1 promoted TDP-43 phosphorylation enhancing in vivo toxicity [94]. Further evidence suggests that in patients with mutations in the gene *CHMP2B*, encoding a protein critical for autophagy and endosomal trafficking, TDP-43 neurotoxicity is modified through casein kinase-1 [95]. This finding suggests a non-canonical pathway for TDP-43 phosphorylation in patients without *TARDBP* mutations. In an effort to decrease phosphorylation and aggregation, one study utilised a molecule to inhibit casein kinase-1, which successfully prevented TDP-43 phosphorylation and consequently reduced neurotoxicity, which is hypothesised to restore cell cycle and autophagy regulation [96,97]. Further investigations beyond casein kinase-1 are required to elucidate the extent to which phosphorylation of other residues on TDP-43 causes misfolding and aberrant TDP-43 functions.

### 3.3. Alternative Splicing of RNA Targets

Alternative splicing is critical to vary RNA expression and protein production from a single gene. One of the most regarded functions of TDP-43 is its ability to regulate and control alternative splicing of its RNA targets. Dysfunctions in TDP-43 as described above indicate that TDP-43 could lose its function to appropriately splice its mRNA targets. In an extensive RNA array study of TDP-43^Q331K^ mice, 1195 exons were identified to be differentially spliced compared to non-transgenic mice [98]. Several studies have outlined the loss of cryptic intron/exon splicing leading to a loss of function in RNAs associated with neurite outgrowth, RNA metabolism, stress granule formation, and mitochondrial function [98,99]. Neurotrophic receptor sortilin 1 (SORT1) is an essential trafficking regulator of progranulin, a protein vital for neurite outgrowth and development [100]. Under physiological conditions, TDP-43 excludes a premature stop codon in cryptic exon 17b of *SORT1*; however, a loss of nuclear TDP-43 leads to the inclusion of this cryptic exon 17b, leading to a loss of SORT1-progranulin binding and regulation [101]. Similarly in other neurite/cytoskeletal proteins, depletion of nuclear TDP-43 causes the inclusion of exon 14 and 15 in *TNIK* and *SEPT6*, respectively, leading to premature stop codons [102]. Although little is known about *SEPT6*, *TNIK* exon 15 inclusive isoforms are known to differentially regulate cell spreading in cortical neurons [103]. Recent evidence has also established the direct link between TDP-43 and STMN2, a protein critical for neurite outgrowth [84]. In in vitro iPSC-derived motor neurons and STMN transgenic mouse models, disruption of TDP-43 drove the aberrant polyadenylation and splicing of *STMN2* pre-mRNA [84,104,105]. These results solidified the evidence that loss of STMN2 via TDP-43, induces cellular death and neuromuscular junction denervation. Another cause of TDP-43 nuclear depletion includes the impairment of autophagy through cryptic sites on *ATG4B* [106]. In addition to TDP-43 nuclear depletion, regulation of *SMN2* requires the utilisation of TDP43 RRM domains for exon inclusion, else a loss of motor neurons occurs [41].

With regard to RNA metabolism, the depletion of TDP-43 is correlated with the increase in *POLDIP3* mRNA variant 2, a splicing regulator for the mTOR/S6K1 cascade [107]. Additionally, nuclear TDP-43 depletion affects splicing regulation of its targets and other RNA binding proteins. TDP-43 nuclear depletion leads to the inclusion of exon7b in *hnRNPA1B* mRNA, a transcript more prone to aggregation [108]. When this *hnRNPA1B* isoform is expressed in vitro, cytoplasmic inclusions form independently of TDP-43, suggestive of a stress granule RNA binding protein response [108]. These studies are by no means an exhaustive list; however, results demonstrate that mutant TDP-43 or loss of nuclear TDP-43 leads to several alternatively spliced transcripts with unfavourable outcomes. However, not every model has shown similar results, in the TDP-43^M337V^ mouse model some transcript deregulation has been identified, though not enough to ‘serve as a neurodegenerative’ phenotype, suggestive of several RNA dysfunction mechanisms of outside of alternate splicing [109]. Further evidence is required to understand the effect of alternatively spliced transcripts caused by different mechanisms of neuronal toxicity via TDP-43.

### 3.4. TDP-43 Alters Non-Coding RNA Biogenesis and Function

Non-coding RNAs (ncRNAs) have been previously observed as ‘junk’ RNAs, however they in fact contain genetic information which are either processed into smaller products, spliced, or utilised in the regulation of important cellular physiology [110,111,112,113]. The functions of ncRNAs appear to be growing as new classes, subunits, or structures are discovered. For example, ncRNAs play a role in chromatin maintenance [114,115], ribosomal interactions [116], gene silencing [117], and many other functions [118,119]. Although there are many classes of ncRNAs, recent evidence has shed light on the possibility that microRNAs (miRNAs) and long non-coding RNAs (lncRNAs) are dysregulated or play some role in the aetiology of ALS [113,120,121].

miRNAs are small ncRNAs of 20–22nt which regulate gene expression by binding to the 3′UTR of an mRNA to repress translation [122,123]. Previous evidence has proposed miRNAs as a potential ALS disease biomarker given their expression in cerebrospinal fluid [124,125], patient plasma [126,127], and patient spinal cord tissues [128]. However, it is still unknown how, which, and why miRNA dysfunction occurs in ALS. The maturation of a miRNA is a complex process, where primary (pri-miRNA) and precursor miRNAs form complexes with RNA binding proteins to facilitate its regulatory functions [25]. In particular, TDP-43 and the Drosha DGCR8 complex is formed to regulate pri-miRNA processing [25]. Subsequent knockdown of TDP-43 in in vitro models lead to significant reductions of specific miRNAs in the nucleus and cytoplasm, demonstrating the importance of TDP-43 and miRNA biogenesis [25]. On the other hand, overexpression of TDP-43^WT^ or TDP-43^ΔNLS^ interferes with proper miRNA biogenesis [129]. Specifically, there were cytoplasmic TDP-43 significantly altered 65 miRNAs compared to 25 significantly altered miRNAs due to nuclear TDP-43, demonstrating a negative role of cytoplasmic TDP-43 on miRNA biogenesis and perhaps implication in the disease mechanism [129]. The research available suggests a role a neuroprotective role of miRNAs, and further understanding of its dysregulation is warranted [129].

lncRNAs are a class of transcripts defined as 200nt or longer [130]. lncRNAs have been implicated in a variety of functions including pluripotency [131], cell cycle regulation [132], stress granule formation [133], epigenetic status [134], and retrotransposon silencing [135]. Research have implicated the binding between TDP-43 and lncRNAs [38,136]. Given the various poly-adenylation sites of TDP-43, they may act as lncRNA promoters as seen in other RNA binding protein and ncRNA complexes [137]. *NEAT1* is a lncRNA which forms paraspekles in response to stress, there has been evidence suggesting colocalization of TDP-43 to paraspeckles in ALS patient motor neurons [138]. Some evidence suggests that TDP-43 alters polyadenylation of *NEAT1* to induce paraspeckle formation [139]. TDP-43 has a strong binding affinity to the full-length isoform *NEAT1_1*, any loss of physiological TDP-43 leads to the upregulation of the shorter isoform *NEAT1_2* [140]. In a drosophila model, upregulation of *NEAT1_1* is neuroprotective and ameliorates TDP-43 proteinopathy [141]. The mechanism by which TDP-43 binds to *NEAT1* is perhaps promoted by RRM1 positive regulatory actions, whereas RRM2 plays a role in tRNA mediated suppression, however this is yet to be confirmed in other models [138,142].

Other lncRNAs such as transposable elements have also been linked with ALS. Upon deletion of a transposable element in the gene *Malat1*, marked levels of unfolded protein response and DNA damage due to TDP-43 localisation occurred [143]. This demonstrates the importance of transposable elements promoting nuclear retention and decreasing interaction sites for TDP-43 aggregation [143]. Another study showed evidenced of the polyadenylated lncRNA *gadd7* binding to TDP-43, however this interaction and downstream mechanisms are still to be elucidated [144]. Although the relationship between ALS and ncRNAs is still in its infancy, a great amount of evidence suggests ncRNAs as a dysregulated pathway in RNA metabolism in ALS. The relationship between ncRNAs and other functions of RNA metabolism including splicing, organelle function, cellular communication, and epigenetics should warrant further investigation.

### 3.5. TDP-43 and Epigenetic Regulation

Epigenetic modifications (excluding phosphorylation) are a relatively understudied area of RNA metabolism in ALS. There are two main classes of epigenetic modifications which regulate chromatin structure and DNA accessibility: histone tail modifications and DNA modifications [145]. The most characterised epigenetic modification is methylation, the addition of a methyl group to a dinucleotide CpG [146,147]. DNA methylation is mostly known to suppress transcription of ‘unwanted’ or ‘excess’ genes [145]. However, in ALS the methylation of critical genes may spur the progression of disease [148,149]. In ALS it is not known what exactly causes epigenetic modifications, however it has been proposed that oxidative stress or mutant genes causes DNA methylation as described in post-mortem ALS spinal cords [148].

Given the involvement of TDP-43 in gene regulation, recent studies have investigated its relationship with epigenetic modification. The main question stands as a ‘chicken and egg’ problem; does TDP-43 alter epigenetic modifications of other genes, or does epigenetic modifications occur on TDP-43? It has been proposed that epigenetic modifications of the *TARDBP* promoter is responsible for changes in TDP-43 protein in healthy aged mouse tissues, however no evidence has been put forth for any model of ALS [26]. Currently, evidence points towards TDP-43 as a modulator of epigenetic modifications for other genes. In a drosophila model, TDP-43 impaired the recruitment of the chaperone Chd1 to chromatin, inducing stress granule formation [150]. Another study has demonstrated that TDP-43^WT^ overexpression in an in vitro SH-SY5Y model lead to the upregulation of the heterochromatin marker, H3K9Me3, a known repressor of transcription [151]. In the same model, the TDP-43^M377V^ mutant resulted in reduction of H3S10Ph-K14Ac a chromatin modifier and mitotic marker [151]. Other modifications such as acetylation was also identified in this TDP-43 model, where binding of histone deacetylase 1 (HDAC1) was facilitated via TDP-43 RRMs [152]. It has been previously demonstrated that acetylation via TDP-43 induces TDP-43 pathology in muscle cells [153]. By inhibiting binding of TDP-43 to HDAC1, TDP-43 induced cell death was ameliorated [152]. These studies suggest a detrimental role of mutant TDP-43 in histone modifications, however the extent still remain unknown [152].

To date, no study has investigated the implication of epigenetic modifications in *TARDBP* specific mutants in either patient derived in vitro models or post-mortem tissues. Peculiarly, in post-mortem spinal cords a significant reduction in methylation and hydroxymethylation in cells that lacked healthy nuclear TDP-43 was observed [154]. The authors suggested that methylation is an earlier sign of disease, and thus not observed by the time TDP-43 pathology and cell death occurred. However, this result stands in stark contrast to previous results demonstrating significantly increased levels of methylation in a wide variety of (non-TDP-43 specific) models of ALS such as: DNA samples [155,156,157,158], post-mortem tissues [128,148,159], various SOD1 rodent models [160], and in vitro iPSCs [161]. Although not an exhaustive list of potential epigenetic modifications, few studies have shed light on the specific involvement of TDP-43 in epigenetic modification. What evidence has been gathered thus far sheds light on a budding area of opportunity to further our understanding of this disease. Interestingly, given the reversibility of many epigenetic modifications, this area stands as an understudied area for therapeutic target investigations for ALS.

## 4. Oxidative Stress and Stress Granule Formation

Oxidative stress and stress granule formation occurs when the antioxidant defence against free radicals and reactive oxygen species fails. In post-mortem ALS tissues, widespread damage induced by oxidative stress is apparent in all central nervous system tissues [162]. The first indication of oxidative stress in ALS was the link between mutant SOD1 protein and disease, where the clearance of reactive oxygen species was impaired [163,164]. The second indication of oxidative stress and stress granules as a disease mechanism are the accumulation of stress granule cytoplasmic foci that form independent of proteinopathies such as TDP-43 in ALS [165]. The third indication includes altered RNA metabolism in non-coding RNAs such as *NEAT1*, as described above [140]. Stress granules consist of polyadenylated mRNAs, translation initiation factors, small ribosomal subunits, and RNA-binding proteins [166]. Under physiological conditions, stress granules are formed as a transient measure against stress by serving as a temporary site of RNA processing such as nonsense mediated decay. Overall, the state of stress granule assembly has been shown to be dependent on oxidative stress, and perhaps an early indicator of cytoplasmic inclusion formation.

In ALS however, the formation of stress granules occurring directly due to TDP-43 is still unknown. In some cases, the presence of TDP-43^A315T, Q331K, or Q343R^ mutations have enhanced stress granule formation [62]. Some studies have described the co-localisation of TDP-43 with stress granule components such as G3BP1 [167], TIA-1 [168], TBC1D1 [169], and even TDP-43 RRMs [47]. In other cases, the effect of TDP-43 on targets such as the *hnRNPA1B* isoform is described as a key component of stress granule formation [170]. For example, in drosophila, the TDP-43^G298S^ mutation results in loss of the microtubule gene *futsch* (human *MAP1B*), and the subsequent degraded transcript colocalises with stress granules [171]. To recapitulate and better understand the relationship between TDP-43 and stress granule formation, several groups have demonstrated that under stress, TDP-43 is recruited to already formed stress granules [47,62,172]. Importantly, it was predominantly cytoplasmic TDP-43 mutants which were recruited to stress granules instead of healthy full length TDP-43 [62]. However, in in vitro models under prolonged states of oxidative stress or injury, loss of nuclear TDP-43 was apparent due to an increase of TDP-43 export with minimal nuclear return of TDP-43 [173]. This loss of nuclear TDP-43 was attenuated when the oxidative stress resistance protein 1 (*OXR1*) gene was upregulated; this occurred alongside a decrease of stress granule formation [173]. Nevertheless, in light of evidence suggesting that stress granules can form independently of TDP-43 [47,62,174], an understanding of the timing at which TDP-43 is pulled into stress granules would be beneficial for further delineating whether this is a mechanism through which TDP-43 can drive disease pathology.

## 5. Mitochondrial Dysfunction

### 5.1. Mitochondrial Structure

Healthy mitochondria are critical in maintaining cellular metabolism and homeostasis. The mitochondrion is a double membraned organelle. The outer mitochondrial membrane separates the mitochondria from the cytoplasm and facilitates the passage of crucial molecules into the inter-membrane space. The inner mitochondrial membrane separates the inter-membrane space with the matrix, where it is home to the oxidative phosphorylation (OXPHOS) machinery, playing a crucial role in facilitating ATP production. The membranes are home to hundreds of proteins which work together to maintain calcium homeostasis, autophagy, and protein folding with other mitochondrion and organelles [175]. Previous findings have identified that TDP-43 contains three mitochondrial localisation domains, denoted as M1, M3, and M5; M1 (aa35–41) resides within the NLS, M3 (aa146–150) within RRM1, and M5 (aa294–300) within the CTD [176]. This leads to the possibility that mutant TDP-43 could bind to mitochondria in place of healthy TDP-43, therefore hindering the normal metabolic functions TDP-43 exerts on mitochondrial proteins (Figure 2).

Structural abnormalities of mitochondria have been reproduced in several ALS TDP-43 models; fibroblasts transfected with TDP–43^G298S^ or TDP-43^A382T^ [176], NSC-34 motor neuron like cells transfected with TDP-43^Q331K^, TDP-43^M337V^, or with TDP-43-C terminal fragments [61,177,178], HEK293T transfected with TDP-43^G298S^ or TDP-43^A382T^ [176], primary mouse motor neurons transfected with TDP-43^Q331K^ or TDP-43^M337V^ [177], and transgenic TDP-43^A315T^ or TDP-43-C terminal fragment mouse models [179,180,181,182]. At the most basic level, abnormal mitochondrial morphology including vacuoles and decreased cristae exhibit markers of distress and mitophagy [179]. Post-mortem studies have reported localisation of mutant TDP-43 to ‘swollen’ mitochondria in ventral horn motor neurons of ALS patients [183]. This finding has been corroborated in other post-mortem studies where cytoplasmic TDP-43 was shown to localise to the mitochondria, in the absence of colocalisation with other structures such as the endoplasmic reticulum, golgi apparatus, lysosomes, and endosomes [9,176]. In other studies however, TDP-43 induced mitochondrial death has been attributed to the p62-autophagy pathway [61] and a novel pathway known as mitoautophagy [184].

Abnormal mitochondria would exhibit mitophagy, however, further evidence is required to elucidate which specific mitochondrial function(s) are disrupted by TDP-43. Reports suggest that TDP-43 can only enter the mitochondria through the outer mitochondrial membrane and inner mitochondrial membrane via TOMM20/70 and TIMM22, respectively [176,185]. However, in a TDP-43 HeLa model, co-localisation between transfected mutant TDP-43 and TOMM20 was not observed [186]. Interestingly in an in vitro TDP-43-KO model and a TDP-43^P497S^ murine model, a significant reduction in TOMM20 and TIMM44 was observed, however the impact of this reduction remains unknown (Figure 2) [187]. In an NSC-34 model, an estimated 13% of truncated TDP-43 has been shown to localise to the mitochondrial matrix, where levels of wild-type TDP-43 never exceed 6% [188]. The M1 localisation signal is most likely responsible for regulating the localisation of mutant and wild-type TDP-43 to mitochondria, given that the abolishment of M1 inhibits TDP-43 localisation and reduces neuronal toxicity [176]. However, further evidence is required to understand the effect of truncated TDP-43 in the mitochondrial matrix, and to elucidate its role in mitochondrial metabolism.

### 5.2. OXPHOS Complex Impairment

Oxidative stress leads to several pathways of mitochondrial damage, in particular altering mitochondrial metabolism via OXPHOS complexes. Given the ability for truncated and wild-type TDP-43 to localise to the mitochondrial matrix, the residency of OXPHOS, it is plausible that mutant TDP-43 could exert detrimental effects on OXPHOS. Indeed, overexpression of TDP-43^WT^, TDP-43^G298S^, TDP-43^A315T^, TDP-43^Q311K^, or TDP-43^M337V^ in NSC34 or HEK293T cells leads to a significant reduction in overall ATP production and specifically, a reduction in complex I activity (Figure 2) [9,176,178,188]. Complex IV was affected slightly in these models, however complex II, III, and V were all unaffected in HEK293T TDP-43^A315T^ and NSC-34 TDP-43^Q331K^ or TDP-43^M337V^ models [9,178,188]. Unfortunately, these in vitro findings have not been replicated in transgenic TDP-43^A315T^ mice nor in fibroblasts derived from patients carrying the TDP-43^A382T^ mutation [189,190]. However, in fibroblasts derived from sporadic ALS patients, ATP production is significantly decreased, and there is a lack of respiratory reserve due to a reduction of complex I (Figure 2) [191].

It is reasonable to posit that perhaps some TDP-43 mutations affect the binding of TDP-43 to mitochondrial complexes, while mutations that cause TDP-43 C-terminal fragments could underpin localisation of TDP-43 to the mitochondrial matrix to disrupt mitochondrial DNA (mtDNA) via the mitochondrial localisation motifs (Figure 2) [176]. Under stressed conditions, complex I is thought to be the major site of reactive oxygen species production, leading to a flux of mtDNA damage, membrane permeability, and structure alterations [192,193]. Congruent with this, overexpressed TDP-43 aberrantly affects mtDNA translation for complex I subunits, resulting in a reduction in complex I (Figure 2) [188]. Specifically, mutant TDP-43 binds to the mRNA encoding two subunits (ND3 and ND6) of complex I to impair their expression, causing disassembly [176]. Taken together, these findings suggest that mutant TDP-43 docks to complex I and/or complex I mtDNA thereby disrupting complex I translation and mitochondrial metabolic function. Overall, the growing body of evidence highlighting links between TDP-43 and mitochondria sets the precedent for further investigations that are focused on understanding the full impact of TDP-43 on mitochondrial metabolism.

### 5.3. Disrupted Mitochondrial Pathways and Functions

The effect of mutant TDP-43 on mitochondria extends beyond OXPHOS dysfunction. Reduced membrane potential [9,61,176], reduced oxygen consumption rate [176], increased reactive oxygen species, and susceptibility to oxidative stress have been observed in in vitro TDP-43 mutant models (Figure 2) [9,61,176]. It is suggested that mitochondrial stress quality control proteins LonP1 [194] and FOXO3a [195] are activated as response to stress, causing TDP-43 mislocalisation to the cytoplasm. However, other mitochondrial related proteins are also affected, leading to mitochondrial dysfunction and organelle-organelle miscommunication. The TDP-43^Q331K^ mouse model has been shown to have significant downregulation of NEK1, which regulates cell death and mitochondrial membrane permeability via VDAC1 phosphorylation (Figure 2) [196,197]. Interestingly, overexpression of TDP-43 leads to the significant reduction of VDAC1, an outer mitochondrial membrane protein responsible for mediating calcium signalling with the Grp75-IP3R complex [61,198]. A loss of VDAC1 leads to a shift away from mitochondrial localisation to co-localisation with TDP-43 and Hsp60 (a facilitator of protein folding) in the cytoplasm [182]. A reduction of NEK1 and VDAC1 could possibly lead to a defective unfolded protein response, loss of membrane potential, decreased NADH oxidation, and increased apoptosis [199]. Other theories suggest that mitochondrial function and calcium signalling is mediated by TDP-43 induced GSK3β activation, leading to a decrease in VAPB-PTPIP51 interactions at the mitochondria and endoplasmic reticulum complex (Figure 2) [186]. While these reports indicate that there is a role of TDP-43 in affecting mitochondrial proteins involved in organelle-organelle communication, specifically for calcium signalling and the unfolded protein response, additional evidence is needed to elucidate the functions and downstream effects of TDP-43 on these mitochondrial proteins involved in organelle communication.

### 5.4. Mitochondrial Dynamics

An important (and often overlooked) capability of mitochondria are their innate ability to undergo fission and fusion with one another—known as mitochondrial dynamics. This unique capability allows mitochondria to maintain cellular networks, metabolise ATP, and ensure cells have enough healthy mitochondria [200,201]. Mitochondrial dynamics are delicately controlled by a variety of fusion and fission proteins. Fusion of the outer mitochondrial membrane is regulated by the large GTPases; mitofusion 1 (MFN1) and mitofusion 2 (MFN2), whereas fusion of the inner mitochondrial membrane is regulated by optic atrophy 1 (OPA1) [202]. Fission dynamics are controlled by the large cytoplasmic dynamin related protein (DNML1/DRP1) and is activated upon recruitment to the mitochondria and facilitated by mitochondrial fission factor (MFF), fission 1 (FIS1), and MiD48/51 [203,204]. Mitochondrial dynamics are known to govern mitochondrial size, shape, location, and homogeneity of mitochondria. Fission is known to eliminate defective mitochondria as a protective agent for the cell by inducing autophagy, however excessive fission events lead to mitochondrial fragmentation and cell death [205,206]. Any impairment of or mutations in these fission and fusion factors could ultimately lead to impaired mitochondrial functioning, localisation, and a loss of cellular functions. The exact steps which dictate mitochondria fusion and fission are still unknown, however research suggests the aid of actin filaments and the endoplasmic reticulum for normal mitochondrial dynamics [207,208,209].

A number of in vitro and in vivo studies have begun to uncover a link between TDP-43 and altered mitochondrial dynamics. In a live in vivo mitochondrial imaging study of TDP-43^A315T^ mice, abnormal retrograde movement of mitochondria is observed prior to the onset of disease [180]. This is followed by mitochondrial anterograde movement abnormalities in the latter stages of disease, and followed by defects in mitochondrial morphology [180]. Unsurprisingly, decreased expression of MFN1 alongside an increased expression of DRP1 and FIS1 has also observed in these mice (Figure 2) [179]. In neurons with TDP-43 cytoplasmic inclusions, significant mitochondrial fragmentation is observed in the axons, as well as increased fission events compared to fusion events [177]. Upon expression of the TDP-43^Q331K^ mutation in primary neurons, DRP1 is dephosphorylated, leading to altered dynamics and neuronal toxicity (Figure 2) [210]. In TDP-43 patient derived fibroblasts, an apparent decrease in mitochondrial interconnectivity and mitochondrial elongation of ~50% is observed alongside a 3-fold increase in DRP1 protein [211]. Efforts to rescue defects in mitochondrial dynamics emphasise that abnormal TDP-43 localisation to either the cytoplasm or within mitochondria causes negative impacts on mitochondrial fusion:fission ratios. For example, overexpression of *MFN2* in rat TDP-43^M337V^ derived primary motor neurons rescues fusion and subsequent trafficking defects [177], while inhibition of TDP-43 localisation to mitochondria results in an overall improvement in mitochondrial function, with mitochondrial dynamics potentially returning to homeostasis [176].

With a wide range of disrupted mitochondrial functions caused by TDP-43, a further understanding of TDP-43-mitochondrial interactions is required to elucidate the extent and breadth of pathological consequences in ALS. This is of particular relevance given evidence of TDP-43 eliciting inflammation via mitochondria [185] and reports of bioenergetic phenotypes in ALS [212].

## 6. Mitochondria, Energy Metabolism, and TDP-43 in ALS

To meet the large energy demands of neurons in the central nervous system, glucose is utilised as the predominant energy substrate to produce ATP via mitochondrial OXPHOS [213]. For simplicity, we consider the two main pathways which generate ATP via the citric acid cycle: glycolysis and β-oxidation. Glycolysis is the metabolic pathway where glucose is converted to pyruvate, then acetyl-CoA to generate ATP via OXPHOS. Whereas β-oxidation catalyses fatty acid molecules to generate acetyl-CoA, NADH, and FADH_2_ which are then utilised in OXPHOS. Despite β-oxidation producing more energy per mole compared to glycolysis, β-oxidation is limited by glucose oxidation through channel import inhibition [214], whereas the rate of glycolysis is dependent only on the availability of glucose in the plasma. Although β-oxidation may seem preferential, the process generates excessive amounts of reactive oxygen species, which would be detrimental to a vulnerable cell.

In ALS, an overwhelming amount of evidence suggests that energy metabolism is perturbed leading to dysregulation of cellular [215,216,217] and whole-body energy homeostasis [218,219,220,221,222,223]. Theories as to the cause for motor neuron vulnerability in the context of metabolic dyshomeostasis primarily relate to alterations in glucose and lipid (fatty acid) metabolism in ALS patients [220], post-mortem patient tissues [224] and patient derived iPSC-motor neurons [216]. One theory is that in an effort to meet energy demands in a dysregulated motor neuron with dysfunctional mitochondria, a switch to fatty acid oxidation could occur—however could cause severe detrimental effects due to an increase in reactive oxygen species. Indeed, a switch from glucose oxidation towards fatty acid oxidation has been demonstrated in a SOD1^G93A^ mouse model [220,225,226]. One mechanism whereby a switch from glucose substrate utilisation to fatty acid utilisation is through the upregulation of UCP2, a mitochondrial uncoupling protein [227]. Interestingly, in NSC-34 TDP-43^Q331K^ and TDP-43^M337V^ models, mutant TDP-43 directly increased the expression of UCP2, however no indication of its effect on glucose or fatty acid oxidation was described [178]. Given the link between TDP-43, mitochondria, substrate utilisation, and cellular/whole-body metabolism, below we explore the possibility that TDP-43 is one potential driver of metabolic phenotypes in ALS.

### 6.1. Mouse Models

Despite the plethora of TDP-43 mice available for modelling ALS, evidence to demonstrate links between TDP-43 and changes in cellular glucose/lipid or systemic metabolism is scarce. No studies to date have utilised TDP-43 derived mouse tissues for in vitro analysis of metabolomics or lipodomics. However, at the systemic level, perhaps the most readily available data pertains to weight loss, which is a common phenomenon in ALS that is often used as a proxy to inform of whole-body metabolic balance. In TDP-43 mice, the timing of the onset of weight loss varies greatly, ranging anywhere from seven days of age in *TARDBP*-inducible KO mice [228], to 35 days of age in amiR-TDP-43i mice [229], to 90 days (male)-180 days (female) in TDP-43^A315T^ [230], to 420 days of age in TDP-43^A315T^ mice [231], to 700 days in TDP-43-CKO mice [232], and not until end-stage or death in TDP-43^N390D^ mice [233]. While the range of mechanisms that underpin weight loss in TDP-43 mice remain to be elucidated, it is likely due to a number of factors including altered food intake [230,234], a loss of muscle mass [181], and a loss of fat mass [228]. In support of TDP-43 playing a potential role in regulating systemic energy metabolism, a 7-fold reduction in *Tbc1d1*, an essential gene required for facilitating glucose uptake in skeletal muscles and fat deposition has been observed in *TARDBP* mice [228,235]. Moreover, the murine model that overexpresses TDP-43^A315T^ exhibits increased body fat percentage, decreased lean mass percentage, but no difference in blood glucose, cholesterol, or triglycerides between compared to their control counterparts [169]. The increase in body fat was explained through the upregulation of *Tbc1d1* mRNA in skeletal muscle, rather than an increase in caloric consumption [169]. By contrast, in TDP-43^Q331K^ mice, no alterations in *Tbc1d1* mRNA was observed, despite these mice losing over 40% of their hind limb muscle mass [236]. Although limited, current data suggest that some *TARDBP* mutations may alter *Tbc1d1* mRNA, subsequently altering body composition and systemic metabolism through a switch to fatty acid oxidation due to a loss of glucose uptake. Extensive studies are needed to determine whether TDP-43 contributes to metabolic dyshomeostasis in ALS, and whether this might be underpinned by TDP-43-mitochondrial interactions.

### 6.2. Human Studies

Many studies have investigated metabolism in ALS patient-derived cell, and in sALS patients. Patient-derived cell models could prove to be a useful tool to investigate ALS disease mechanisms given the ability to capture similar ALS-like cell death pathways and TDP-43 mislocalisation as a hallmark of disease progression [237,238]. Investigating energy metabolism in patient fibroblasts, one group found a downregulation of NADH metabolism correlating with disease progression [239]. In another fibroblast study, downregulation of glucose and lipid metabolic genes led to perturbed glucose and fatty acid metabolism [240]. Similar findings suggested that ~25% of sALS patients had perturbed mRNAs encoding proteins relevant for metabolism; these perturbed mRNAs led to accelerated glucose metabolism, oxidative stress, and eventually cell death [241]. This pathway of accelerated glucose metabolism and oxidative stress suggests either perturbed glycolysis and/or potentially a switch to fatty acid β-oxidation. In *C9orf72* patient fibroblasts the levels of oxidative stress in lipids was increased 2-fold leading to autophagy [242]. In other models such as patient derived myotubes, increased fatty acid oxidation was observed [220]. This change in substrate utilisation could be explained by a downregulation of *PGC-1α*, which would drive a switch from glycolysis to β-oxidation [243]. In a multi-omics analysis of ALS iPSC-derived motor neurons, markers of lipid metabolism were enriched in a cohort of 17 patients [216]. In particular the loss of arachidonic acid biosynthesis was identified as a potential cause of lipid dysregulation particularly in the *SOD^A4V^*, *C9orf72, TDP-43^Q343R^*, and sALS lines [216]. Taken together these studies provide in vitro, in vivo, and ex vivo evidence of dysregulated lipid metabolism pathways in ALS.

In ALS patients hypermetabolism, hyperlipidemia, decreased body mass index and lower glucose use in the brain are associated with survival [219,244,245,246,247,248,249,250,251,252]. However, human studies investigating metabolic alterations specifically in the context of TDP-43 do not exist. Although *TARDBP* mutations are extremely rare in ALS patients, TDP-43 aggregates and signalling could explain some of the observed alterations in systemic metabolism—as seen in in vitro studies as stated above. There are several potential mechanisms through which TDP-43 can act on metabolic pathways. It is proposed that post-translational modifications of lipid metabolism pathways can alter patient phenotypes [157], or mediate expression of genes such as *SREBP2* a mediator of cholesterol metabolism [253]. TDP-43 is known to dysregulate LXR signalling, proteins specifically encoded to absorb and transport cholesterol, leading to an excess of cholesterol [254,255]. Interestingly the LXR signalling pathway is known to be regulated by ncRNAs, suggestive of a link between RNA metabolism and lipid metabolism [256]. Perturbed clearance of this excess cholesterol is demonstrated in the downregulation of *CYP27A1*, in patients with TDP-43 proteinopathy [257]. Additionally in a genomic study, patients with higher circulating lipids, had genetic variants in *HNRNPK* [258], a known cytoplasmic aggregator of TDP-43 [259]. Further explanation of dysregulated RNA and lipid metabolism is demonstrated through the complex binding of lncLSTR-FXR-apoC2; when lncLSTR-FXR-apoC2 is depleted, increased triglyceride clearance and lipoprotein lipase is activated [181]. In cerebrospinal fluid studies, ALS patients display a significantly different lipodomic profiles [260] with alterations in ceramides and glucosylceramides [261] homocysteines [262], and cholesterol [263]; these studies suggest a hypercatabolic state and decreased cholesterol clearing leading to neuronal tissue injury in patients [264]. Despite the presence of TDP-43 pathology in the vast majority of ALS patients, no study to date have thoroughly investigated the direct effects of *TARDBP* mutations on systemic metabolism in patients. Thus, this an exciting area of research that remains to be explored.

## 7. Discussion and Future Directions

ALS is a multifaceted and complex neurodegenerative disorder. The heterogeneous nature of this disease presents an enormous challenge in understanding the mechanisms and identifying efficient therapeutic targets. Early theories of ALS mechanisms surrounding proteinopathies and oxidative stress have been largely explored, however only until recently have they been considered mutually inclusive. TDP-43 cytoplasmic inclusions are a hallmark of ALS, and they are likely a disease target which unites several plausible mechanisms of disease. Each domain of TDP-43 proves essential in its respective role, however little outside of the CTD and the subsequently generated C-terminal fragments have been explored. Some evidence has demonstrated that the alteration of either the NLS, NES, or RRM domains facilitate cytoplasmic aggregation and RNA metabolic defects. Unfortunately, the exact mechanisms on how these aggregates form in patients or models without *TARDBP* mutations or TDP-43 overexpression remain unknown. With the unique ability of TDP-43 to bind and alter the expression of thousands of RNA and DNA targets, there is little argument in discounting potential effects-whether direct or indirect-in altering cellular metabolism. Based on the evidence available, it appears that oxidative stress is an early indicator of cellular dysfunction. It also appears that the combination of oxidative stress and aggregated TDP-43 exerts a negative impact on mitochondrial function and downstream cellular metabolism. In surveying the evidence to date, we posit that the combination of cellular stress and TDP-43 aggregates leads to a relentless feedback loop between oxidative stress, stress granule formation, TDP-43 cytoplasmic aggregation, downregulation of essential mitochondrial and metabolic proteins, altered mitochondrial function and dynamics, a switch to fatty acid β-oxidation, and altered cellular metabolism—which collectively compounds leading to motor neuron death (Figure 3).

With a plethora of evidence suggesting several cellular and mitochondrial defects in ALS, it is highly unlikely that one therapeutic would be able to repair or attenuate facets leading to motor neuron toxicity. To better understand the effect of TDP-43 on aberrant RNA function, mitochondrial dysfunction, and cellular and systemic metabolism, more relevant TDP-43 human iPSC-derived motor neuron and/or organoid models might prove to be beneficial in facilitating translational research outcomes in ALS. Such translation would be strengthened by complementary studies in humans.

## Figures and Tables

**Figure 1 metabolites-12-00709-f001:**
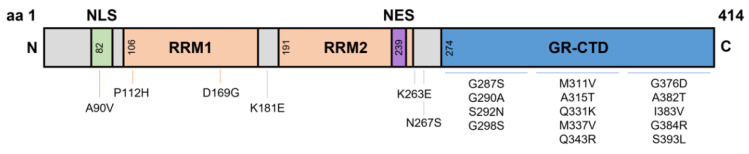
Domains of TDP-43 and known Single Nucleotide Polymorphism mutations. TDP-43 is a 43 kDa protein containing 414 amino acids. The NLS resides from aa82–98, with RRM1 and RRM2 residing at aa106–176 and aa191–259, respectively. The NES resides within RRM2 at aa239–250, and a GR-CTD is located at aa274–414. Most known single nucleotide polymorphisms reside on the GR-CTD. However, some mutations have been identified at the NLS and around the RRMs. aa, amino acid; NLS, nuclear localisation signal domain; RRM, RNA recognition motifs; NES, nuclear export signal; GR-CTD, glycine rich C-terminal domain.

**Figure 2 metabolites-12-00709-f002:**
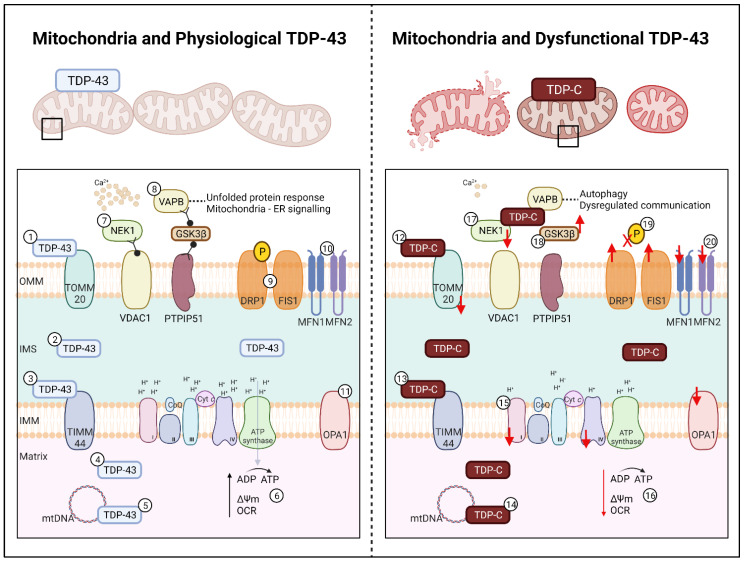
Effect of TDP-43 on Mitochondria in ALS. **Left**: Mitochondria and Physiological TDP-43. In healthy cells length TDP-43 localises to (1) TOMM20 on the OMM, (2) within the IMS and to (3) TIMM44 of the IMM, (4) freely within the matrix, and (5) regulates mtDNA for OXPHOS complex expression. (6) OXPHOS complexes I-IV and ATP synthase facilitate ATP production, facilitating the maintenance of the membrane potential and a healthy oxygen consumption rate. (7) Mitochondrial proteins NEK1 and VDAC1 support calcium signalling. (8) The mitochondrial-endoplasmic reticulum complex, PTPIP51-GSK3β-VAPB ensures proper protein folding and organelle communication. The mitochondrial dynamics related proteins for (9) fission on the OMM (DRP1, FIS1) and (10) fusion on the OMM (MFN1, MFN2) and the (11) IMM (OPA1) facilitate mitochondrial network homeostasis and regulation. **Right**: Mitochondria and Dysfunctional TDP-43. In ALS, fragmented TDP of 25 kDa and 35 kDa (denoted as TDP-C) localises to (12) TOMM20 and (13) TIMM44 causing downregulation and loss of mitochondria. TDP-C also co-localise to (14) mtDNA causing fragmentation, leading to a (15) loss of OXPHOS complex I and IV. (16) Loss of complex regulation leads to a decrease in ATP production, driving a loss of membrane potential and oxygen consumption rate. (17) TDP-C leads to a decrease in NEK1, causing a loss of VDAC1 and calcium signalling. (18) Localisation of TDP-C to GSK3β causes the loss of mitochondrial-endoplasmic reticulum signalling resulting in protein folding dysfunction and loss of organelle communication, which in turn leads to autophagy. Further impacts of TDP-C results in (19) DRP1 de-phosphorylation leading to (20) increased fission and decreased fusion. Upregulation of FIS1 and downregulation of MFN1, MFN2, and OPA1 are also observed in TDP-43 models leading to increased fission and mitochondrial fragmentation and subsequent cellular death. TDP-C; TDP-43 C-terminal fragments, OMM; outer mitochondrial membrane, IMS; inter-membrane space, IMM; inner mitochondrial matrix, ATP; adenosine triphosphate, CoQ; Coenzyme Q; Cyt *c*; cytochrome *c*. Created with BioRender.com accessed on 12 July 2022.

**Figure 3 metabolites-12-00709-f003:**
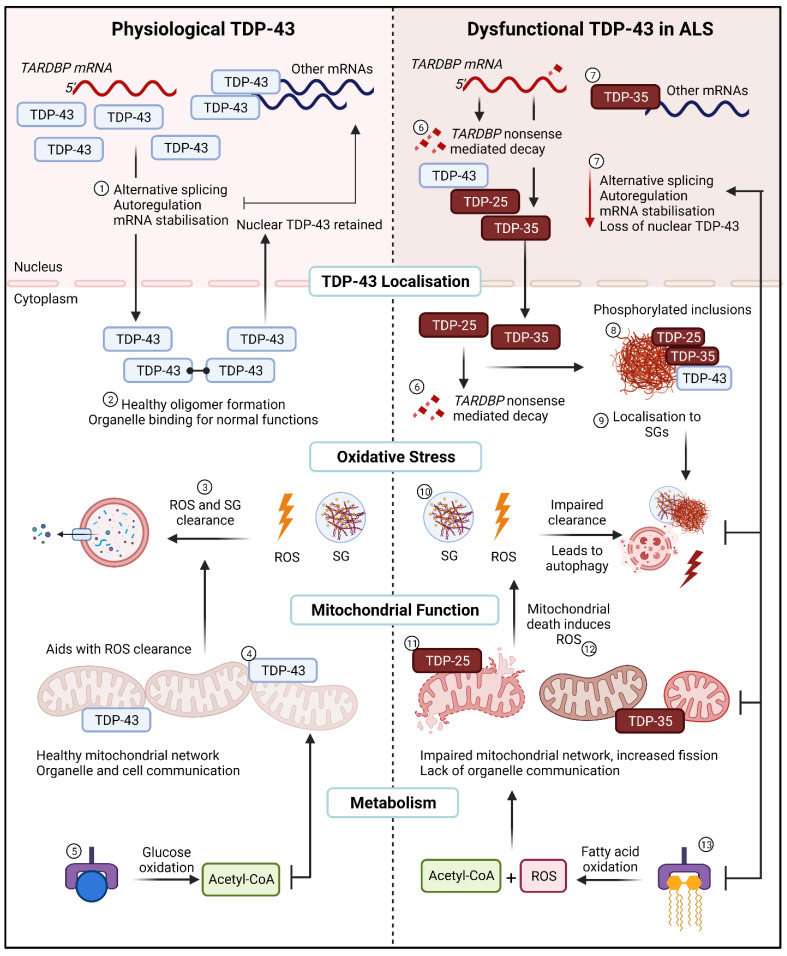
The relentless TDP-43 positive feedback loop in ALS. (**Left**), Physiological TDP-43. (1) In healthy cells TDP-43 regulates normal RNA metabolic functions such as alternative splicing, autoregulation, and mRNA stabilisation. (2) TDP-43 oligomers are generated in the cytoplasm and cleared upon return to the nucleus. (3) Under normal oxidative stress conditions, ROS and SG are cleared by the aid of healthy mitochondria and a lack of mutant TDP-43 aggregating with SGs. (4) Full-length TDP-43 binds to the mitochondria to undergo normal functions and aids in organelle and cell communication. (5) Glucose as the main substrate for energy utilisation for ATP in mitochondria is utilised. (**Right**), Dysfunctional TDP-43 in ALS. (6) Dysfunctional TDP-43 transcribes and translates fragmented TDP-43, some fragmented are sent for nonsense mediated decay in the nucleus or the cytoplasm. (7) A loss of RNA metabolic functions is observed alongside a loss of nuclear TDP-43. (8) TDP-43 and TDP-43 fragments localise to the cytoplasm, where they are phosphorylated and form aggregates. (9) These aggregates lead to sequestration into SGs. (10) Impaired clearance of ROS and SG results in autophagy and cell death through a lack of apoptosis. (11) Mutant TDP-43 bind to mitochondria to impair the mitochondrial network, as well as mitochondrial signalling and function. (12) Lack of mitochondrial function creates a feedback loop with impaired oxidative stress and ROS clearance. (13) Switch from glycolysis to fatty acid oxidation induces ROS alongside Acetyl-CoA for substrate utilisation in mitochondria. SG; stress granules, ROS; reactive oxygen species. Created with BioRender.com accessed on 12 July 2022.

## Data Availability

Not applicable.

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
