# Peer review of "Altered TDP-43 Structure and Function: Key Insights into Aberrant RNA, Mitochondrial, and Cellular and Systemic Metabolism in Amyotrophic Lateral Sclerosis"

_metabolites, 2022, doi:10.3390/metabo12080709_

Round 1

Reviewer 1 Report

Jiang and colleagues discussed how the modulation of cell and RNA metabolism is modulated by TDP-43, an important protein for the development of ALS. Authors associated functional aspects with SNPs and structural alteration occurred in specific domains of TDP-43.

According to me the manuscript is well organized, but the Authors can develop the part associated with non-coding RNAs, TDP-43 and mitochondria. Non-coding RNAs are an important component involved in the regulation of cell physiology and pathology. For example, the Authors discussed that TDP-43 has different poly-adenylation sites. Can this affect the light of 3-UTR and therefore the binding with miRNAs? Moreover, TDP-43 promotes miRNA biogenesis (https://doi.org/10.1073/pnas.1112427109) or binds miRNAs that can affect ALS progression (https://doi.org/10.3389/fncel.2020.00117). The importance of non-coding RNAs in the modulation of TDP-43 is demonstrated by the fact that NEAT1 can ameliorate TDP-43 toxicity (DOI: 10.1080/15476286.2020.1860580 ).

Reviewer 2 Report

This review summaries the roles of TDP-43 in ALS. It first describes the structure of TDP-43 and the function of each TDP-43 domain. It then develops how TDP-43 is regulated, its role in RNA processing, in oxidative stress and stress granule formation, in mitochondrial regulation, function and metabolism.

The review is clearly written and well organised. The authors made an effort to summaries key points in figures. There is a good coverage of the literature as well.

Only very minor points to consider below:

- page 2 line 75, problem with reference insertion in the text

- figure 1 should be referred to in the main text 

Reviewer 3 Report

The review article titled “Altered Tdp-43 Structure and Function: Key Insights into Aberrant RNA, Mitochondrial, and Cellular and Systemic Metabolism in Amyotrophic Lateral Sclerosis “by Leanne Jiang and Shyuan T Ngo describes about the role of TDP-43 in various aspects of ALS pathology. The authors have systematically surveyed the literature and designed the article in a way that encompasses important aspects of ALS that are affected by TDP-43 structure and function. The manuscript is nicely written and easy to follow. There are few minor suggestions that might improve the overall impact of the manuscript:

1.       The authors could include latest publications about Stathmin transgenic mice. There are few references about Stathmin and I think authors can increase the number of these references.

2.       TDP-43 has important role in epigenetic modifications (methylation/hydroxymethylation). The authors could include this information to make the manuscript more appealing.

3.       Please correct reference/style in line 75, and 104.    

Round 2

Reviewer 1 Report

The authors responded to all my questions and included information on non-coding RNAs in the manuscript.